# Does Far-Infrared Therapy Improve Peritoneal Function and Reduce Recurrent Peritonitis in Peritoneal Dialysis Patients?

**DOI:** 10.3390/jcm11061624

**Published:** 2022-03-15

**Authors:** Yuanmay Chang, Jui-Ting Chang, Mei-Yi Lee, Mei-Zen Huang, Yann-Fen C. C. Chao, Yung-Luen Shih, Yao-Rong Hwang

**Affiliations:** 1Institute of Long-Term Care, MacKay Medical College, New Taipei 25245, Taiwan; hwangyr@mmc.edu.tw; 2Division of Nephrology, Department of Internal Medicine, Shin Kong Wu Ho-Su Memorial Hospital, Taipei 11101, Taiwan; skhnephropgy@gmail.com; 3Department of Physical Therapy, Fooyin University, Kaohsiung 83102, Taiwan; michelin54321@hotmail.com; 4Department of Nursing, National Tainan Junior College of Nursing, Tainan 70007, Taiwan; meizen@mail.ntin.edu.tw; 5General Education Center, Cheng Shiu University, Kaohsiung 83301, Taiwan; yannfen.chao@gmail.com; 6Laboratory Medicine, Department of Pathology, Shin Kong Wu Ho-Su Memorial Hospital, Taipei 11101, Taiwan; t005524@ms.skh.org.tw

**Keywords:** end-stage renal disease, peritoneal dialysis, far-infrared therapy (FIR)

## Abstract

The use of peritoneal dialysis in end-stage renal disease is increasing in clinical practice. The main purpose of this study was to evaluate the effect of far-infrared radiation therapy on inflammation and the cellular immunity of patients undergoing peritoneal dialysis. We recruited 56 patients undergoing peritoneal dialysis, and we included 32 patients for the experimental group and 24 patients from the control group in the final analysis. The experimental evaluation in our study was as follows: (1) We used abdominal computed tomography to explore the changes in abdominal blood vessels. (2) We compared the effects of peritoneal dialysis using blood glucose, HbAlC, albumin, urea nitrogen, creatinine, white blood cells, hs-CRP; peritoneal Kt/V of peritoneal function, and eGFR. (3) We compared the cytokines’ concentrations in the two groups while controlling for the other cytokines. Results and Discussion: (1) There was no significant difference in the abdominal blood vessels of the experimental group relative to the control group according to abdominal CT over the 6 months. (2) Our study demonstrates statistically significant effects of FIR therapy on the following parameters: creatinine (*p* = 0.039 *) and hs-CRP (*p* < 0.001 **) levels decreased significantly, and eGFR (*p* = 0.043 *), glucose (*p* < 0.001 **), and albumin (*p* = 0.048 *) levels increased significantly. Our study found that in the experimental group, creatinine and hs-CRP levels decreased significantly due to FIR therapy for 6 months. However, our study also found that the glucose level was significantly different after FIR therapy for 6 months. Peritoneal dialysis combined with FIR can reduce the side effects of the glucose in the dialysis buffer, which interferes with peritoneal inflammation and peritoneal mesothelial cell fibrosis. (3) In addition, we also found that no statistically significant difference in any inflammatory cytokine after FIR therapy. IFN-γ (*p* = 0.124), IL-12p70 (*p* = 0.093), IL-18 (*p* = 0.213), and TNF-α (*p* = 0.254) did not exhibit significant improvements after peritoneal dialysis with FIR treatment over 6 months. Conclusions: We found that the effectiveness of peritoneal dialysis was improved significantly with FIR therapy, and significant improvements in the peritoneal permeability and inflammatory response were observed.

## 1. Introduction

The use of peritoneal dialysis (PD) in end-stage renal disease (ESRD) is increasing in clinical practice. In Taiwan, diabetes is the main cause of kidney disease, and as many as 40% of dialysis patients suffer from nephropathy due to diabetes [1,2]. Relevant studies have confirmed that PD is the favorable treatment for end-stage renal disease. However, the biggest limitation of this treatment is that as PD—with a high dextrose 4.25% PD solution—continues, the peritoneum denatures, resulting in diffusion and convective transport, causing losses in ultrafiltration ability [3]. In clinical studies, it is reported that the main reasons for PD failure are complications due to peritoneal inflammation and mesothelial cell destruction, which result in conditions such as long-term PD uremia, repeated peritoneal infections, dialysis buffering, low pH, and sugar decomposition poisoning [4,5]. Moreover, these factors can also cause interstitial cell degeneration and fibrosis, fibrous deposition, vascular proliferation, the output of plasma coagulation factor, and fibrin infiltration into the abdominal cavity, thereby reducing the dialysis ultrafiltration rate [6].

Glucose is the most commonly used osmotic agent in PD solutions. The continuous absorption of dextrose leads to hyperglycemia, exacerbation of diabetes mellitus (DM), hyperlipidemia, obesity, malnutrition, and titration of insulin dose in DM patients. Hyperglycemia can cause changes to the renal cell structure and glomeruli increase, which induces glomerular proliferation, glomerular basement membrane thickening, and fibrosis [7]. Glucose degradation products (GDPs) contribute to reduced dialysis function and the pathogenesis of peritoneal membrane fibrosis [8,9,10]. A previous study showed that long-term exposure to non-biocompatible high-concentration glucose dialysate and repeated peritonitis are the main causes of changes in the structure and function of the peritoneum [4,11,12]. During an acute inflammatory process such as peritonitis, natural killer (NK) cells produce inflammatory cytokines such as tumor necrosis factor alpha (TNF-α) and interleukin (IL-6). Moreover, through production of IFN-gamma (IFN-γ) and transforming growth factor-β1 (TGF-β1), these cells may directly orchestrate the fibrotic process. Interleukin-l beta (IL-1β), IL-6, and TNF-α are proinflammatory hormones which trigger a persistent inflammatory response in the peritoneum, ultimately leading to structural changes in the peritoneum [13,14,15,16].

Far-infrared rays have been widely used in clinical healthcare and rehabilitation since 1800. Certain studies found far-infrared can vibrate the water molecules in our body to produce energy [17,18]. Far-infrared rays are called biological waves, and the effects produced are called biological effects [19,20]. The energy produced is divided into two categories: thermal and non-thermal, according to the form the energy takes [21]. FIR was able to increase skin blood flow using thermal therapy in mice and rats [22,23]. Akasaki et al. (2006) found that FIR promotes angiogenesis in mice through a non-thermal effect; and that FIR inhibited IL-6 and TNF-α activity in mice with peritonitis [24]. In recent years, more studies confirmed and elucidated the effects of far-infrared rays in the human body. For example, FIR has been shown to promote blood circulation [23,25], relieve fatigue and pain [26,27], and promote wound healing [25,28,29,30].

In regard to the non-thermal effects, studies show that far-infrared rays can promote cell proliferation and wound healing [31,32], and can increase the expression of arterial endothelial nitric oxide synthase and nitric oxide synthesis in hamsters [33]. Generation of new blood vessels from angiogenesis is accompanied in all states by increased vascular permeability. Far-infrared radiation therapy can promote hemeoxygenase-1 (HO-1) and endothelial nitric oxide synthase (eNOS) generated by the L-arginine/nitric oxide pathway, reducing inflammation through the inhibition of intimal hyperplasia and reducing oxidation. It can even stimulate the inhibition of TNF-α to produce anti-inflammatory effects [23,29,32,34,35,36,37]. Although the mechanisms involved in FIR are not clear, various studies indicate that far-infrared rays can promote the proliferation of bone marrow stem cells and keratinocytes through the CXCR4/ERK or Notch1/Twist pathways [31,38,39,40]. Excessive oxidative stress and inflammation are the major factors that promote abnormal cell proliferation. Lin et al. (2013) found that far-infrared rays can enhance the activity of vascular endothelial cells and promote angiogenesis [41] to increase permeability for material exchange between tissue and blood vessels [42], and can inhibit inflammation by inhibiting inflammatory factors, such as IL-6 and TNF-α [24,43].

The literature concerning the effect of far-infrared radiation therapy on inflammation in patients undergoing PD is limited. Therefore, in this study, we aimed to study the physiological condition and inflammatory response of dialysis patients who had been irradiated with far-infrared rays for 6 months. The main purpose of this study was to evaluate the effects of far-infrared radiation therapy on inflammation and the innate immunity cytokines of patients undergoing PD.

## 2. Materials and Methods

### 2.1. Research Objects

In this study, we selected patients with the following conditions from the PD area of the Shin Kong Wu Ho-Su Memorial Hospital medical center, Taipei. The baseline blood samples were collected via the median cubital vein the day before the experiment began. We aimed to (1) perform abdominal computed tomography to assess mesenteric vascular stiffness; (2) obtain gender, duration of peritoneal dialysis, body weight/body mass index, blood glucose level, glycated hemoglobin (HbAlC) level, albumin concentration, urea nitrogen (BUN) concentration, creatinine concentration, peritoneal Kt/V of peritoneal function, the estimated glomerular filtration rate (eGFR), white blood cell (WBC) concentration, and high-sensitivity C-reactive protein (hs-CRP) concentration; (3) analyze the innate immunity cytokines released after PD patients had been irradiated with FIR, including IL-1 β, IFN-γ, IL-4, IL-6, IL-12p70, IL-18, and TNF-α, in order to confirm the effects of the inflammatory response in PD patients.

### 2.2. Patient Selection

We conducted this experimental clinical trial from January to October 2016 in the PD unit of a medical center in northern Taiwan. There were 151 patients on PD in the medical center, and 71 of these met the conditions for the trial (Figure 1). The patients were randomly assigned to the control and experimental groups by their medical numbers assigned by the hospital.

We obtained informed consent from all participants.

Inclusion criteria were:(1)Patients with PD stabilized for 6 months and attending daily routine PD.(2)No peritonitis at least 6 months, with 2.5% PD fluid, with an acceptable plasma creatinine ratio (D/P Cr) and peritoneal permeability, no need to use hypertonic PD fluid, and a C-reactive protein (CRP) index that did not indicate infection.(3)Patients under the age of 75 years who could perform PD fluid exchange alone or with the assistance of a family member.(4)FIR taken at least 4 times a week.

Exclusion criteria were:(1)Patients who underwent kidney transplantation or were hospitalized during the study period.(2)An inability to perform a complete far-infrared radiation study for 6 months.

We conducted a power analysis for the repeated-measures analysis of variance test using G*Power version 3.1.2 to determine the sample size, setting the power to 0.8, the effect size to 0.20, and the significance level to 5% (two-tailed). Based on a calculation with the group allocation ratio of 1:1, 52 patients (26 in each group) were considered necessary to achieve the desired power and effect size. We assumed an annual mortality rate of 10%, and the drop-out rate for the study reached 25%. Hence, we needed at least 35 patients in each group.

### 2.3. Experimental Method

Before going to bed, the PD patients had their fluids emptied and replaced each day; they were then irradiated for 40 min after the fluid input. We conducted monthly patient visits and kept records of FIR exposure time collected by researchers. We drew blood from the patients every third month, and the data from the 6-month period were assessed.

### 2.4. Experimental Disposal (FIR Instrument)

This study used the WS TY301 FIR Emitter (WS Far-Infrared Medical Technology, Taipei, Taiwan) far-infrared instrument, which has a wavelength range of 3–25 μm, and a peak of 8 μm [44,45]. The treatment was centered on the navel region, i.e., the entire abdominal cavity, from the lower edge of the sternum to the groin. The recommended safe distance that still allows reaching the skin was 25 cm [44,45]. Before going to bed, the PD patients had their fluids emptied and replaced each day. They were then irradiated for 40 min using the far-infrared instrument (WS TY301 FIR emitter) after the fluid input (Figure 2). The patients in the other group did not receive far-infrared radiation treatment and served as the control group. The dialysis of patients in the control group was compared with that of the experimental group after irradiating the abdomen with far-infrared rays for 6 months.

### 2.5. Centrifugal Collection of Plasma

Blood samples were also collected via the median cubital vein every three months for routine blood checks. The sample collecting tubes contained ethylene diamine tetraacetic acid (EDTA) as an anticoagulant, so that the blood could be separated into plasma and blood cells by centrifugation for further experiments.

(1)Plasma collection: The complete blood cell count and differential count were acquired (CBC and DC). Blood collection tubes contained EDTA, and the plasma was centrifuged at 800× *g* for 15 min. The upper supernatant of the plasma was pipetted equally into 110 μL Eppendorf tubes. The remaining white blood cells in the Buffy coat were collected into another tube, and hemolysis was performed using erythrocyte lysis buffer to remove erythrocytes without cell nuclei. A 0.4 mL solution with white blood cells was separated equally into 200 μL Eppendorf tubes to collect RNA and DNA.(2)Determination of immune factors in plasma: The plasma, peritoneal dialysate, and associated supernatant were used for analysis of cytokines using Human Th1/Th2 Panels of Luminex 200 (BioRad Corporation, Madison, WI, USA). We analyzed innate immunity cytokines, including IL-1β, IL-4, IL-6, IL-12p70, IL-18, IFN-γ, and TNF-α.

### 2.6. Experimental Evaluation

The study used abdominal Computed Tomography (CT) to measure the degree of stiffness of the mesenteric vessels in order to explore the changes in abdominal blood vessels. The checkpoints were as follows: before and after the FIR therapy for 6 months. According to the CT scans at the two checkpoints, it was based on mesenteric atherosclerotic plaque, lumen stenosis or occlusion, and morphological changes in bowel and mesenteric ischemia. We graded the degree of stiffness of the mesenteric vessels on a scale of 0–2, with 0 indicating normal; 1, a faint but detectable change; 2, abnormal [46].The study compared the effects of PD using BUN, creatinine (Cr) value, and the dialysate and plasma creatinine ratio (D/P Cr) from the blood and the peritoneal dialysate (the emptied dialysate fluid) every 3 months.After the blood and peritoneal dialysate were collected, albumin (ALB), BUN, creatinine (Cr), white blood cell (WBC), hs-CRP, and HbAlC (the first and the last time) analyses were used to assess check peritoneal permeability and inflammation.Evaluation of Serum Biochemical Parameters

Plasma was collected in February (before irradiation), May (after irradiation for 3 months), and August (after irradiation for 6 months). The cytokines secreted by type 1 and type 2 helper T cells play important roles in infection. In this study, 7 kinds of cytokines related to type 1 and type 2 helper T cells were screened and analyzed, including IL-1β, IL-4, IL-6, IL-12p70, IL-18, IFN-γ, and TNF-α. We used repeated-measures ANOVA to compare the individual cytokines’ concentrations between the two groups, while controlling for the other cytokines.

### 2.7. Statistical Analysis

The statistical analysis was performed using SPSS statistics (version 26, IBM Singapore Pte Ltd., The IBM Place 486072, Singapore), and standard descriptive statistics were used to assess the baseline characteristics. Data are presented as mean  ±  standard deviation. Repeated-measures ANOVA was performed to compare serial changes in the clinical data and echo parameters. *p*-values < 0.05 reached statistically significant difference.

## 3. Results

### 3.1. Patient Characteristics

We recruited 80 patients undergoing PD in this study. Of these, nine did not meet the inclusion criteria. As described above, we divided the remaining patients into groups according to their assigned numbers: 39 patients were assigned to the experimental group and 32 patients were assigned to the control group. A total of 15 patients dropped out during later parts of the study for various reasons. Two patients died; two patients underwent kidney transplantation; and 11 patients were hospitalized without FIR and withdrew. Hence, we included 32 patients from the experimental group (82.1% of the initial group) and 24 patients from the control group (75% of the initial group) in the final analysis (Figure 1).

The mean age of the patients in the experimental group was 54.53 ± 12.56 years and that of those in the control group was 58.31 ± 8.19 years. In each group, >50% of the participants were female. The mean years of PD of the patients in the experimental group were 11.46 ± 3.54, and those of the control group were 8.76 ± 5.18 (Figure 1). The mean body weight of the patients in the experimental group was 57.63 ± 13.04 Kg, and that of the control group was 63.46 ± 16.41 kg. The mean body mass index (BMI) of the patients in the experimental group was 25.52 ± 4.98 (kg/m^2^), and that of the control group was 24.82 ± 5.55. The mean years of PD of the patients in the experimental group were 11.46 ± 3.54, and those of the control group were 8.76 ± 5.18. Baseline demographic and clinical characteristics of patients who completed the study did not differ significantly between the two groups (Table 2).

### 3.2. Evaluation of Abdominal Blood Vessels

We used abdominal Computed Tomography (CT) to measure the degree of stiffness of the mesenteric vessels. The mean degrees of stiffness of the mesenteric vessels before and after in the experimental group were 0.73 ± 0.16 and 0.72 ± 0.27, and those of the control group were 0.73 ± 0.19 and 0.74 ± 0.25. The results show that there was no significant difference in the degrees of stiffness of the mesenteric vessels (*p* = 0.256) in the experimental group and in the control group (Table 1).

### 3.3. Physiological Evaluation

Using blood samples, we assessed all physiological and biochemical responses after baseline and 6 months of FIR intervention. In Table 2, it can be observed that the baseline demographic and clinical parameters between the two groups of patients were similar, except for glucose (*p* < 0.001 **), creatinine (*p* = 0.039 *), albumin (*p* = 0.048 *), eGFR (*p* = 0.043 *), and hs-CRP (*p* < 0.001 **) levels. No significant difference in BUN concentration (*p* = 0.121) or WBC count (*p* = 0.365) was found between the experimental group and the control group. We observed that the BUN concentration increased in the control group and decreased in the experimental group over the 6 months.

No significant difference in the peritoneal Kt/V (0.097) or weekly CCr (*p* = 0.173) of peritoneal function was found between the experimental group and the control group. We observed that the Kt/V increased in the experimental group and decreased in the control group over the 6 months. We also observed that peritoneal weekly CCr decreased in the experimental group and increased in the control group over the 6 months. Even though no significant difference in the BUN concentration (*p* = 0.121) or WBC count (*p* = 0.365) was found between the experimental group and the control group, we observed that the BUN concentration increased in the control group and decreased in the experimental group over the 6 months. We also observed that the WBC count decreased in the experimental group and increased in the control group over the 6 months.

### 3.4. Evaluation of Serum Biochemical Parameters

We noted improvements in the levels of IL-6 (*p* = 0.061), IL-4 (*p* = 0.156), IL-1β (*p* = 0.175), and TNF-α (*p* = 0.254), but these differences did not reach statistical significance (Table 3 and Figure 3). However, the plasma IL-18 levels in the experimental group and the control group were higher than those in the other two observation periods at the third month (*p* = 0.213) (Table 3 and Figure 3). A higher level of IL-12p70 was detected in the plasma of patients in the experimental group at the third month, which lasted until the sixth month (*p* = 0.093) (Table 3 and Figure 3). However, the plasma IFN-γ of the experimental group decreased in the sixth month, reaching a lower value than in the third month (*p* = 0.124) (Table 3 and Figure 3). Although repeated-measures ANOVA showed that the pre–post values improved in the experimental group as compared to the control group, the differences were not statistically significant (Table 3). Repeated-measures ANOVA also revealed inhibition of IL-6 and TNF-α and increased IL-12p70, but these differences did not reach statistical significance (Table 3 and Figure 3).

## 4. Discussion

### 4.1. Abdominal Vessels

Diagnosis is highly accurate based on mesenteric atherosclerotic plaque, lumen stenosis or occlusion, and morphological changes in bowel and mesenteric ischemia. However, the abdominal vessels of our patients may have had no or non-specific manifestations in the early stage, and CT diagnosis is difficult and requires comprehensive clinical manifestations. The results are similar to those reported by Ren [47]. Encapsulating peritoneal sclerosis (EPS) is an inflammatory condition. Most studies have consistently identified increasing PD duration as a key risk factor for the development of EPS. Other parameters that may be possible risk factors for EPS include blood sugar, higher dialysate glucose exposure, use of conventional PD solutions, peritonitis (frequent, severe, or prolonged), ultrafiltration (UF) failure, and a high peritoneal solute transport rate (PSTR) [48]. According to the research, the average duration of peritoneal dialysis infection is about 5 years [49,50]. The average duration of dialysis in our study was more than 8 years, and the average HBA1C of the cases themselves was over 9%. However, it should be considered that our study was controlled by inclusive and exclusive criteria. It is the effect of EPS on the peritoneal efficiency which should be considered a minimal bias in this study.

Peritonitis is an inevitable complication in PD cases, and most peritonitis is caused by intestinal bacterial translocation or fluid contamination. We observed that blood WBC, hs-CRP, frequency of peritonitis infection, and the bacterial species could influence the peritoneal efficiency and permeability. These are factors that deserve further study.

### 4.2. The Physiological Effects of PD Due to FIR

Concerning peritoneal function and serum biochemical parameters post-FIR therapy, the statistically significant effects of FIR therapy were on the following parameters: (1) creatinine (*p* = 0.039 *) and hs-CRP (*p* < 0.001 **) levels decreased significantly, and (2) eGFR (*p* = 0.043 *), glucose (*p* < 0.001 **), and albumin (*p* = 0.048 *) levels increased significantly.

We found a small increase in albumin in both groups from the third month to the sixth month. Peritoneal dialysis patients should pay attention to the loss of protein in the peritoneum. When the serum albumin concentration is less than 4.0 g/dL, the mortality risk is increased. During the study, there was guidance and there were provisions for enhancing nutritional intake of high biological value protein, so it was found that the albumin in this study increased slightly. It should be noted in the study that the increase in albumin would also have increased the changes in other physiological values of PD. It should be considered to have been a minimal confounder in this study.

Our study found that, in the experimental group, creatinine decreased significantly due to FIR therapy for 6 months. Creatinine is often used to assess the severity of renal insufficiency. In the experimental group, creatinine decreased significantly, and the glomerular filtration rate increased significantly, indicating that far-infrared irradiation can improve the effect of PD, which is in accordance with the literature [17,41]. As the marker of dialysis adequacy, we also checked BUN, the peritoneal Kt/V, and weekly CCr of peritoneal function post-FIR therapy. Despite no statistical significance in the changes in BUN, peritoneal Kt/V, and weekly CCr of peritoneal function between the two groups, we observed a good change in PD for FIR. Our study found that the BUN, creatinine, and peritoneal weekly CCr decreased in the experimental group. The peritoneal Kt/V and eGFR of peritoneal function increased in the experimental group, which means that the effect of PD increased with FIR. However, urea nitrogen was accumulated in the blood without being excreted, resulting in an increase in the degree of renal function damage in the control group. We expected FIR to increase the peritoneal permeability and slow down the rate of peritoneal degeneration. The effects of FIR therapy of decreasing creatinine, increasing eGFR, and improving dialysis adequacy are worth evaluating in future studies to confirm the results in this study.

The average duration of peritoneal dialysis, glucose concentration, HBA1C, and BMI in our study were more than 8 years, around 100 mg/dL, 9%, and 25 kg/m^2^. These results are similar to those of Sowinski et al. (2020) and Mehrotra (2017), who reported that PD patients absorb 60% of dextrose in each exchange. The continuous PD absorption of dextrose leads to hyperglycemia and obesity in diabetic PD patients [8,9]. Future studies should be conducted to confirm the effects of FIR therapy on higher dextrose dialysate and the comorbidities of PD patients, and the influence of the patient’s blood glucose level.

Our study also found that, in the experimental group, the glucose concentration (*p* < 0.001 *) increased significantly due to FIR therapy for 6 months. Long-term exposure to high-concentration glucose dialysate and repeated peritonitis are the main causes of changes in the structure and function of the peritoneum [4]. The glucose concentrations in different dialysis buffers may impact the burden of kidney filtration rate [11]. Studies have shown that the glucose concentration also promotes peritoneal inflammation and the production of transforming growth factor by peritoneal mesothelial cells, finally leading to fibrosis [12]. Therefore, PD combined with FIR can reduce the side effects of glucose in the dialysis buffer, which interferes with peritoneal inflammation and peritoneal mesothelial cell fibrosis. This finding is in accordance with the literature [11].

The thermal effects of FIR treatment were vasodilation and increased vascular flow [17]. Lin et al. (2013) found that far-infrared rays can enhance the activity of vascular endothelial cells and promote angiogenesis [41] to increase permeability for material exchange between tissue and blood vessels [42], and increase peritoneal transport and solute removal clearance. It was also significantly improved after far-infrared radiation [51]. Therefore, we found that the effects of PD were significantly improved with FIR therapy.

In addition, our study also found that hs-CRP in the experimental group decreased significantly due to FIR therapy for 6 months. hs-CRP acute reactive protein is a serum protein in the acute phase, which can be used as a good indicator of local inflammation or tissue injury [52]. Despite no statistical significance being observed in WBC count (*p* = 0.365) between the experimental group and the control group, we observed that the WBC count decreased in the experimental group and increased in the control group over the 6 months. Our results demonstrate that patients in the experimental group undergoing FIR therapy had significantly reduced inflammatory responses, and may have exhibited improved vascular endothelial cell function. Chang (2018) and Lee et al. (2019) also mentioned that FIR treatment can improve vascular endothelial cell function and help reduce the inflammatory response [24,28]. Accordingly, we found that patients in our experimental group showed significant improvements in the inflammatory response [53].

### 4.3. Inflammatory Cytokines

In Table 3, it can be seen that no statistical differences in the concentrations of inflammatory cytokines with/without FIR therapy were observed in our study. Originally, we hoped to see a similar situation to that revealed by Chang (2018) as regards IL-6/TNF-α/INF- γ [24]. In the present study, IFN-γ (*p* = 0.124), IL-12p70 (*p* = 0.093), IL-18 (*p* = 0.213), and TNF-α (*p* = 0.254) did not exhibit significant improvements in PD with FIR treatment for 6 months. However, as cytokines such as IL-6, TNF-α, and INF-γ are short-term inflammatory cytokines, we believe that changes in IFN-γ, IL-12, IL-18, and TNF-α can be classified as long-term inflammatory changes, according to the inflammatory situation in this study. TNF-α is mainly related to systemic inflammation [53]. As the TNF-α in the plasma of the subjects in the experimental group continuously decreased in the three observation periods, it was confirmed that the clinical inflammatory symptoms of the subjects improved due to the influence of far-infrared radiation treatment, like the physiological phenomenon in our study (i.e., hs-CRP decreased significantly). However, IFN-γ and IL-18 concentrations in the plasma of patients in the experimental group and the control group were higher at the third month than the first month. IL-18 is considered a pro-inflammatory cytokine. When macrophages are activated by an antigen (such as infection source), they will secrete IL-18, and then chemotactic T cells will secrete IFN-γ. Additionally, IFN-γ stimulates other macrophages or related immune cells to eliminate microbial pathogens [53]. Furthermore, TNF-α and IFN-γ are vital symbols of innate immunity and play important roles in immunotherapy, whereas IL-12 plays critical roles in the activation of innate immunity. IL-12p70 is involved in the process of naive T cells differentiating into type 1 helper T cells, and IL-12 plays critical roles in the activation of innate immunity [53]. High IL-12p70 expression was observed in the plasma of patients in the experimental group at the third month, and this lasted until the sixth month. One of the effects of IL-12p70 is to stimulate the production of IFN-γ, which is anti-inflammatory. However, the IFN-γ in the plasma of the experimental group decreased in the sixth month, having a lower value than in the third month. There was still a certain correlation in the changes of these cytokines as long-term inflammatory. Moreover, in Table 3, one can see the continuous inhibition of IL-6 and TNF-α, and increased IL-12p70 in the experimental group. There were still certain correlations in the changes of these cytokines, indicating inhibition of long-term inflammation by FIR.

### 4.4. Overall Assessment

Peritonitis is an inevitable complication in PD cases, but we hope to delay the inflammatory response through FIR, as used for cases of PD in our study. Although there were no significant differences in the degree of stiffness of abdominal vessels or inflammatory cytokines throughout the study, we found that patients in our experimental group reported improvements in the eGFR and inflammation over the 6 months. FIR therapy can enhance HO-1 synthesis, activate eNOS, inhibit vascular endothelial growth, reduce oxidative vascular sclerosis, and improve the effects of vascular repair [24,34,54] and permeability for material exchange in PD [11]. Chang (2018) and Lee et al. (2019) also mentioned that FIR treatment can reduce and delay IL-6 and TNF-α responses, which may improve vascular endothelial cell function and help reduce the inflammatory response [24,28]. In this study, the effect of FIR on the peritoneum, how it improved the peritoneal exchange material, and how it reduced the incidence of peritonitis in PD patients were only studied over 6 months. It is thought that FIR may slow down the rate of peritoneal degeneration, increase the peritoneal permeability [11], and reduce the frequency of peritoneal inflammation [24]. Therefore, further studies are needed to obtain a better understanding of the effects of FIR therapy on patients with PD treatment.

We observed the feelings and effects of/on the patients when this study was carried out for 6 months from winter to summer. In the winter, the patients felt that the FIR was warm, but in the summer, they felt it was hot. The patients were consulted about the influence of using air-conditioning or electric fans during FIR. It is advisable to control the ambient temperature during FIR.

During the clinical care of patients, we often used examination data to confirm symptom reports. However, using feedback from patients about their experiences with FIR helped us to understand the patients’ feelings and the effects of the treatment [55]. Our patients reported that symptoms of physical comfort corresponded with objective data during the study.

## 5. Limitations

Patient compliance with far-infrared irradiation treatment is the greatest challenge. For some patients from hemodialysis, FIR is routinely provided for A-V fistula site and both feet of patients in this dialysis unit; therefore, patients know the benefits of FIR irradiation. In addition, we had monthly patient visits to track their state and feelings during the execution process, and collected the hours of FIR exposure time from the machine. We ensured compliance with therapy by FIR machine records. FIR should be taken at least four times a week. We excluded two patients whose FIR exposure was less than four times a week.

The biggest limitation of this study is that FIR inhibited the inflammatory response in individual cases. Although it is assumed that the sugar-containing peritoneal dialysate is a pro-inflammatory reactant, it is impossible to use far-infrared irradiation therapy when the inflammatory reaction actually occurs in individual cases. There was no significant relationship between cytokines and inflammatory response data. Changes in cytokines that may be affected by daily exposure to FIR do not necessarily correspond to the inflammatory factors in the blood, which were checked every three months.

Carbohydrate antigen 125 (CA125) is one of the differential diagnostic indexes for peritoneal inflammation. Although it was elevated to varying degrees, the positivity rate was low. CA125 alone cannot be used as the basis for diagnosing inflammation, and it needs to be combined with abdominal tomography and inflammatory factors in the blood for clinical judgment. We did not measure CA125. Further study may collect data on CA 125 as proof of peritoneal inflammation.

## 6. Conclusions

Based on the observed trends, FIR therapy decreases the BUN, creatinine, and perito-neal weekly CCr, increases the peritoneal Kt/V and eGFR of peritoneal function, and im-proves dialysis adequacy. The heat of the FIR treatment caused vasodilation and in-creased vascular flow, increasing permeability for material exchange. There was still a certain correlation among the changes in these cytokines after continuous inhibited IL-6 and TNF-α, and increased IL-12p70 in long-term inflammation by FIR.

## Figures and Tables

**Figure 1 jcm-11-01624-f001:**
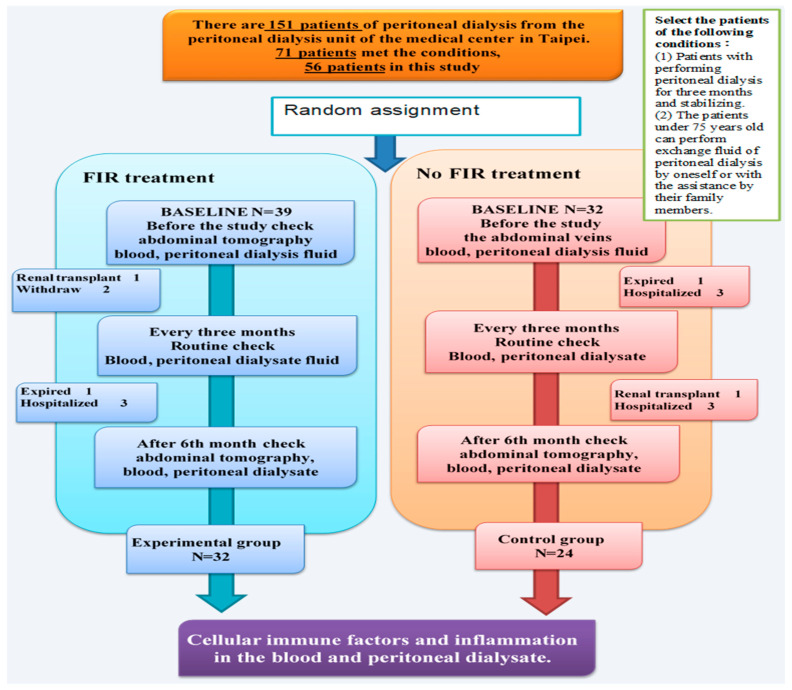
Flow chart of participants comparing the effect of patients undergoing FIR treatment and non-FIR treatment.

**Figure 2 jcm-11-01624-f002:**
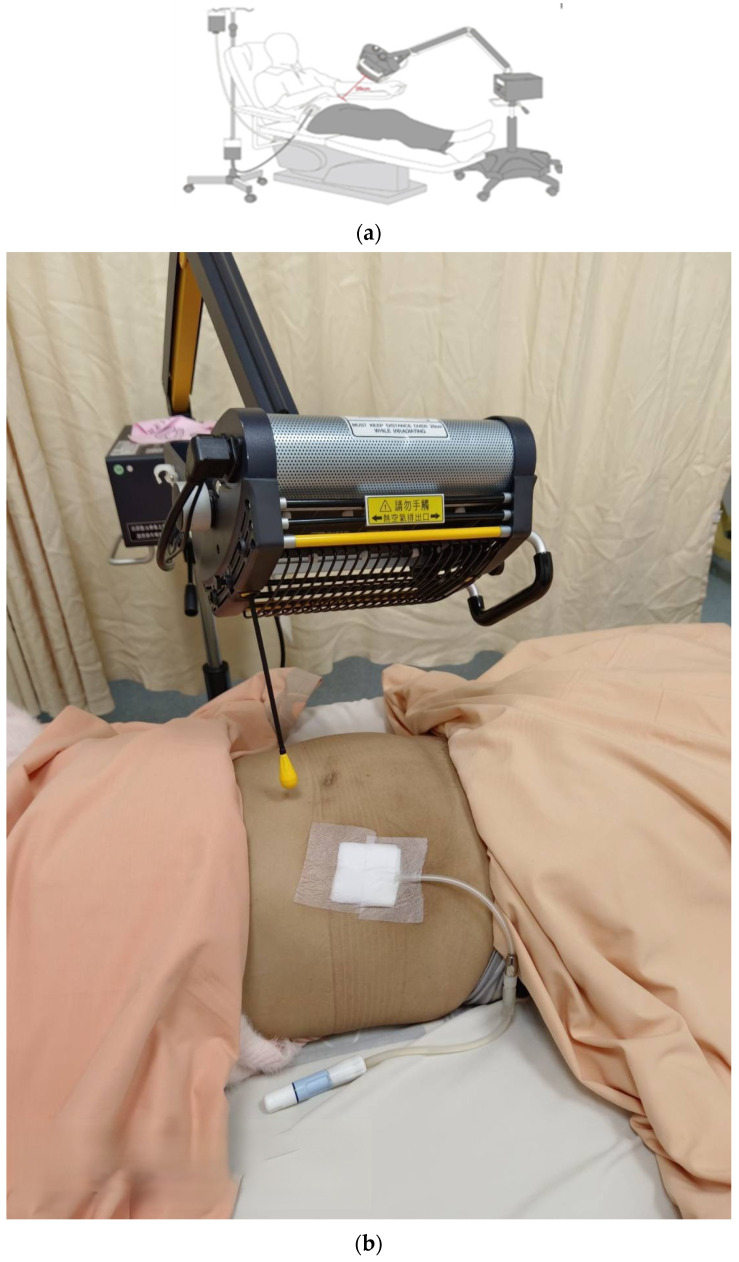
(**a**) FIR therapy set-up during a PD exchange. Abbreviations: FIR: far-infrared; PD: peritoneal dialysis [11]; (**b**) a photo of the FIR therapy at night.

**Figure 3 jcm-11-01624-f003:**
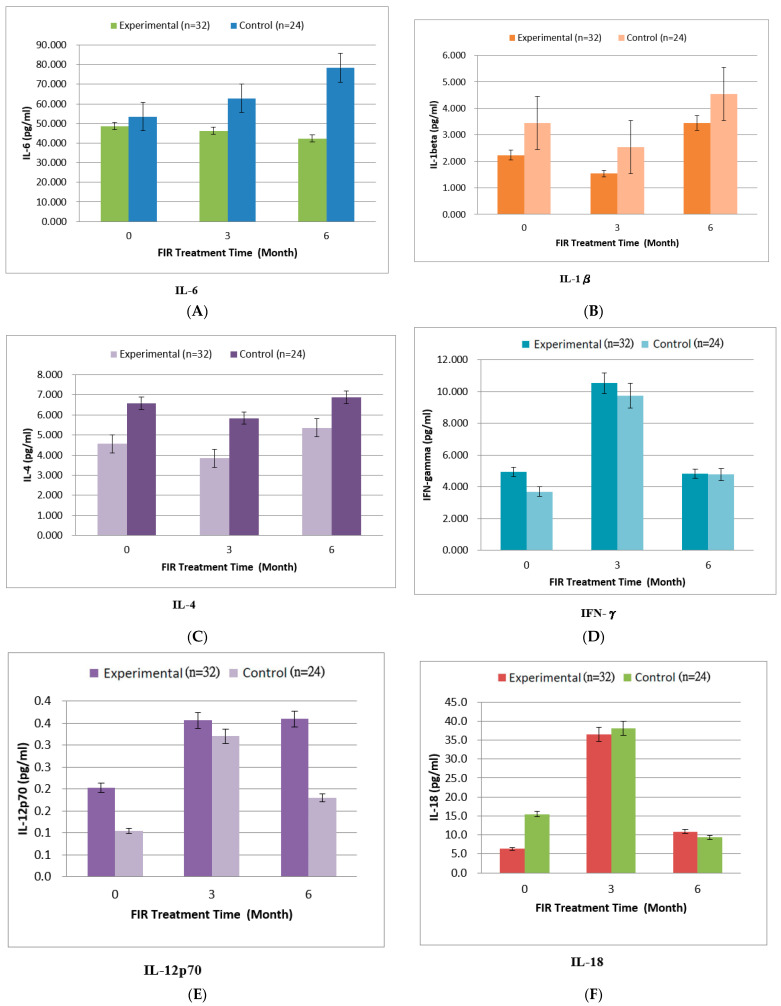
Comparison of clinical biochemical parameters of the experimental group (FIR) and control group (non-FIR). (**A**) IL-6, (**B**) IL-1β, (**C**) IL-4, (**D**) IFN-γ, (**E**) IL-12p70, (**F**) IL-18, (**G**) TNF-α.

**Table 1 jcm-11-01624-t001:** Abdominal blood vessels of the experimental group (FIR patients) and control group (non-FIR patients).

Parameters	Experimental Group (*n* = 32)		Control Group (*n* = 24)		*p*
	0	6 (Months)	0	6 (Months)	
Degree of stiffness (Variation grade)	0.73 ± 0.16	0.72 ± 0.27	0.73 ± 0.19	0.74 ± 0.25	0.256

A scale of 0–2, with 0 indicating normal; 1, a faint but detectable change; 2, abnormal (mesenteric atherosclerotic plaque, lumen stenosis, occlusion, or mesenteric ischemia).

**Table 2 jcm-11-01624-t002:** Baseline demographic and clinical parameters of the experimental group (FIR) and control group (non-FIR).

Parameters	Experimental Group (*n* = 32)			Control Group (*n* = 24)			*p*
	0	3	6	0	3	6	
Age (years)	54.53 ± 12.56			58.31 ± 8.19			0.115
Gender (M/F)	15/17			9/15			0.116
Body weight	57.63 ± 13.04			63.46 ± 16.41			0.425
BMI (Kg/m^2^)	25.52 ± 4.98			24.82 ± 5.55			0.591
Years of PD	11.46 ± 3.54			8.76 ± 5.18			0.253
Peritoneal function							
Peritoneal Kt/V	1.95 ± 0.26	2.03 ± 0.22	2.14 ± 0.17	2.11 ± 0.47	1.95 ± 0.46	1.82 ± 0.58	0.097
Peritoneal weekly CCr (L/week/m^2^)	61.67 ± 14.72	58.24 ± 12.11	56.61 ± 13.28	58.51 ± 15.34	62.14 ± 10.40	65.25 ± 14.41	0.173
Serum biochemistry							
Glucose (mg/dL)	105.27 ± 29.48	111.27 ± 28.37	99.40 ± 12.25	98.85 ± 10.24	99.31 ± 9.75	105.08 ± 13.72	<0.001 **
WBC	6.51 ± 2.26	5.90 ± 1.97	6.68 ± 2.24	8.28 ± 2.74	7.59 ± 1.84	7.89 ± 2.01	0.365
HbA1c (%)	9.93 ± 1.51	9.79 ± 1.65	10.21 ± 1.91	9.68 ± 1.46	10.22 ± 1.69	10.25 ± 1.85	0.757
BUN (mg/dL)	63.67 ± 14.77	62.27 ± 13.03	61.13 ± 12.22	71.85 ± 13.38	64.15 ± 20.40	60.01 ± 12.45	0.42
Creatinine (mg/dL)	9.07 ± 3.48	7.79 ± 4.25	6.23 ± 0.68	12.02 ± 1.69	10.22 ± 1.69	9.16 ± 0.21	0.039 *
Albumin (g/dL)	3.68 ± 0.30	3.57 ± 0.38	3.73 ± 0.43	3.56 ± 0.42	3.67 ± 0.40	3.73 ± 0.56	0.048 *
eGFR	4.01 ± 0.75	3.99 ± 0.73	4.12 ± 0.78	4.21 ± 1.41	3.96 ± 0.91	3.73 ± 0.78	0.043 *
Phosphate (mg/dL)	4.96 ± 1.02	4.83 ± 0.91	5.01 ± 0.68	5.13 ± 1.43	5.02 ± 1.61	5.36 ± 1.12	0.58
T-P	6.52 ± 0.71	6.48 ± 0.54	6.47 ± 0.66	6.44 ± 0.52	6.47 ± 0.58	6.36 ± 0.65	0.314
hs-CRP (mg/dL)	1.31 ± 2.14	0.51 ± 0.54	0.75 ± 1.37	0.98 ± 1.49	1.00 ± 2.21	0.41 ± 0.36	<0.001 **

Continuous variables are presented as mean ± standard deviation. Categorical variables are presented as numbers (percentages). *: *p* < 0.05; **: *p* < 0.01 using the Mann–Whitney U test (two-tailed)/Fisher’s exact test. Kt/V: Kt/V urea; CCr: creatinine clearance; HbA1c: glycated hemoglobin; BUN: blood urea nitrogen; hs-CRP: high-sensitivity C-reactive protein.

**Table 3 jcm-11-01624-t003:** Biochemical evaluation and clinical parameters of the experimental group (FIR) and control group (non-FIR).

Parameters	Experimental Group (*n* = 32)			Control Group (*n* = 24)			*p*
	0	3	6	0	3	6	
IL-1beta	2.23 ± 0.44	1.54 ± 0.14	3.45 ± 0.75	3.45 ± 0.39	2.54 ± 0.42	4.55 ± 1.12	0.175
IL-4	4.56 ± 1.75	3.84 ± 0.87	5.36 ± 2.44	6.58 ± 1.98	5.84 ± 1.58	6.87 ± 2.12	0.156
IL-6	48.6 ± 9.49	46.2 ± 12.35	42.3 ± 11.63	53.5 ± 11.64	58.8 ± 19.42	65.4 ± 21.75	0.061
IFN-γ	4.93 ± 1.44	10.52 ± 2.86	4.83 ± 1.43	3.69 ± 0.94	9.73 ± 2.53	4.78 ± 2.14	0.124
IL-12p70	0.20 ± 0.09	0.36 ± 1.14	0.36 ± 0.09	0.10 ± 0.04	0.32 ± 0.053	0.18 ± 0.07	0.093
TNF-alpha	6.06 ± 2.23	5.07 ± 2.12	4.78 ± 1.47	5.39 ± 2.18	4.61 ± 1.97	5.85 ± 2.11	0.254
IL-18	6.35 ± 2.12	36.48 ± 8.9	10.88 ± 2.47	15.51 ± 4.23	38.11± 11.64	9.34 ± 3.11	0.213

Continuous variables are presented as mean ± standard deviation. Categorical variables are presented as numbers (percentages). Using SPSS statistics (IBM Singapore Pte Ltd., The IBM Place 486072, Singapore), standard descriptive statistics.

## Data Availability

The data presented in this study are available on reasonable request from the corresponding author.

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
