# Peer review of "Does Far-Infrared Therapy Improve Peritoneal Function and Reduce Recurrent Peritonitis in Peritoneal Dialysis Patients?"

_jcm, 2022, doi:10.3390/jcm11061624_

Round 1

Reviewer 1 Report

The idea is novel and interesting. 

Most of my comments and queries are incorporated into the PDF attached. 

  1. The introduction is lengthy and line 60-62 needs clarity e.g how does reduced immunity results in long term PD?
  2. Repetitions need so to be avoided 
  3. Author needs to concisely summarize the potential effects of FIR in other organ systems 
  4. Patient selection in the methods section , first 2 paragraph can be merged. In line 124-126, does the author mean low or slow transporters. Line 126 : CRP that did not increased" does the author mean stable for a period of time or one off reading at the point of recruitment ?
  5. It appears that the pt were selected at one time point and randomly assigned , what method was used to assigned randomly?
  6. What is the date of recruitment if not done at one time point?
  7. Did the author do a power calculation for sample size?
  8. for the patients needing caregiver for dialysis, were the patients able to consent ?
  9. Figure 1 is colorful but information repetitive, can be simplified further. Did all the patients survived the 6 months ? suggest can use outcome numbers  and processes can be parallel alongside 
  10. In the method section , can the author give more details on the dialysis regimen, assuming all patients were on CAPD with night dwells and perform FIR treatment at night , were there APD cases ?
  11. How did the author ensure compliance to therapy ?
  12. Line 172 to 176, the method section does not usually need to provide the aims or reason for doing the test, this should be done in the earlier sections.
  13. Line 162, do not use abbreviations that has not been explained earlier in the article
  14. In the abdominal CT scan that was performed to measure vascular stiffness, can the author elaborate what is being focused on in the abdomen and the pitfalls of this measurement in the discussion. Usually stiffness refers to arteries, can the rationale of using vein as a surrogate ? line 229-231
  15. Did the author take note of volume status of the patients ? (ultrafiltration and urine output.) the latter can decline over 6 months leading to lower GFR, UF if good can be the convective force to remove creatinine , so can fluid overload giving a falsely low creatinine due to dilution.
  16. Line 376-377, what were the clinical improvements that the author was monitoring that was not mentioned in the previous texts, similarly 331-332, what improvements  was reported by the experimental group?
  17. The conclusion should point out the deficiency or limits of the study and the way forward
  18. As the whole procedure is new, if the author can have a photo of how the procedure is done at night would greatly improve the understanding for the reader
  19. How did the author decide on the dose of FIR and the appropriate distance of FIR from skin based on different patient habitus ?
  20. Did the author collected data on CA 125?

Author Response

Thank you for your valuable opinion. We fully agree with your concern.

I revised the manuscript and wrote where changes in line numbers.

Response to Reviewer 1 Comments

Thank you for your valuable opinion. We fully agree with your concern.

Point 1: This work is very interesting, original and probably paving the way to larger experiences. However there are different points of concern.

The most important is the study design. The work included patients’ randomization, thus it should be a randomized clinical trial. If so, there are many methodological aspects that were not observed. For example, a calculation of the most appropriate sample size was not carried out. As concerns randomisation, a crucial point, there is not any detail about it. Was it automated? Which system was used for treatment assignement? Was randomization stratified, for example by dialysis age or other parameters?

Response 1: Please provide your response for Point 1. (in red)

Thank you for your valuable opinion. We fully agree with your suggestion.

We rewrote in Line 125-149.

Point 2: Author needs to concisely summarize the potential effects of FIR in other organ systems

Response 2:

Thank you for your valuable opinion. We fully agree with your suggestion.

We rewrote in Line 48-56, 69-75, 466-470.

Point 3: Patient selection in the methods section, first 2 paragraph can be merged. In line 124-126, does the author mean low or slow transporters. Line 126 : CRP that did not increased" does the author mean stable for a period of time or one off reading at the point of recruitment ?

It appears that the pt were selected at one time point and randomly assigned, what method was used to assigned randomly?

What is the date of recruitment if not done at one time point?

Did the author do a power calculation for sample size?

for the patients needing caregiver for dialysis, were the patients able to consent ?

It appears that the pt were selected at one time point and randomly assigned, what method was used to assigned randomly?

What is the date of recruitment if not done at one time point?

Did the author do a power calculation for sample size?

for the patients needing caregiver for dialysis, were the patients able to consent ?

Response 3:

Thank you for your valuable opinion. We fully agree with your suggestion.

We rewrote in Line 115-149.

Point 4: Figure 1 is colorful but information repetitive, can be simplified further. Did all the patients survived the 6 months? suggest can use outcome numbers and processes can be parallel alongside

Response 4:

Thank you for your valuable opinion. We fully agree with your suggestion.

We changed in Line 150-151.

Point 5: In the method section, can the author give more details on the dialysis regimen, assuming all patients were on CAPD with night dwells and perform FIR treatment at night, were there APD cases?

Response 5:

Thank you for your valuable opinion. We fully agree with your suggestion.

We recruited both CAPD and APD. Before going to bed, the PD patients had their fluid emptied and replaced each day; it was then irradiated for 40 minutes after the fluid input.

Point 6: How did the author ensure compliance to therapy?

Response 6:

Thank you for your valuable opinion. We fully agree with your suggestion.

Patients changed PD fluid every day. We ensured compliance to therapy by FIR machine record. We have monthly patients visited and recorded of FIR exposure time collected by researchers. FIR should be taken at least 4 times a week.

Point 7: Line 172 to 176, the method section does not usually need to provide the aims or reason for doing the test, this should be done in the earlier sections.

Response 7:

Thank you for your valuable opinion. We fully agree with your suggestion.

Please see line 115-122.

Point 8: Line 162, do not use abbreviations that has not been explained earlier in the article.

Response 8:

Thank you for your valuable opinion. We fully agree with your suggestion.

Please see line 178.

Point 9: In the abdominal CT scan that was performed to measure vascular stiffness, can the author elaborate what is being focused on in the abdomen and the pitfalls of this measurement in the discussion.

Usually stiffness refers to arteries, can the rationale of using vein as a surrogate ? line 229-231

Response 9:

Thank you for your valuable opinion. We fully agree with your concern about how measure vascular stiffness.

We consulted a senior radiologist to rechecked the degree of mesenteric sclerosis. Therefore, we revised assessment of the degree of stiffness of mesenteric vessels. We rewrote in Line 242-246, 297-309, and 416-422.

Point 10: Did the author take note of volume status of the patients? (ultrafiltration and urine output.) the latter can decline over 6 months leading to lower GFR, UF if good can be the convective force to remove creatinine, so can fluid overload giving a falsely low creatinine due to dilution.

Response 10:

Thank you for your valuable opinion. We fully agree with your concern about effect of residual renal function (RRF) in initial six months since initiating peritoneal dialysis. In our research, our study was performed during 2016/1 to 2016/10. Reviewing our patients, the shortness duration since receiving peritoneal dialysis to enrolling in this study was more than one year (the latest starting peritoneal dialysis date: 2013/9/2). Therefore, the effect of residual renal function which should be considered had a minimal bias in this study. We have measured Peritoneal Kt/V, Peritoneal weekly CCr, and e GFR in the study. Thank you for your suggestions.

Point 11: Line 376-377, what were the clinical improvements that the author was monitoring that was not mentioned in the previous texts, similarly 331-332, what improvements was reported by the experimental group?

Response 11:

Thank you for your valuable opinion. We fully agree with your concern about Line 331-332, 376-377.

Line 331-332 Chang in 341-351.

The changes in IL-6, IL-12p70, and TNF-α were not statistically significant. However, the changes revealed inhibition in the variety of IL-6, TNF-α, and increased IL-12p70.

in Line 376-377

We also rewrite in the limitation in 431-440.

Point 12: The conclusion should point out the deficiency or limits of the study and the way forward.

Response 12:

Thank you for your valuable opinion. We added in 466-470.

Point 13: As the whole procedure is new, if the author can have a photo of how the procedure is done at night would greatly improve the understanding for the reader

Response 13:

Thank you for your valuable opinion. We added figure 2 from reference 11, also we took photo from a patient at home. We taught the appropriate distance of FIR from skin for the experiment group of participants before this study. Although the different patients have their habitus, FIR machines have safety settings a safe distance. Please see the photo.

Please see line 169-171.

Point 14: How did the author decide on the dose of FIR and the appropriate distance of FIR from skin based on different patient habitus?

Response 14:

Thank you for your valuable opinion. We have written in Line 154-155, 161-162, 431-440.

This study used the WS TY301 FIR Emitter far-infrared instrument, which has a wavelength of 3~25μm, and a peak of 8 μm. The safety distance guard set by the far-infrared instrument and to reach the skin was 25 cm; it was irradiated for 40 minutes using FIR. According to the feeling and effect of the patients, this study was carried out for 6 months from winter to summer. In the winter, the patients felt that the FIR is warm, but the patients felt hot in the summer. The patients have consulted the influence of using air-conditioner or electric fan during FIR.

Point 15: Did the author collect data on CA 125?

Response 4:

Thank you for your valuable opinion. We did not collect CA125 data in our study, however we added a limitation in 459-464.

Reviewer 2 Report

This work is very interesting, original and probably paving the way to larger experiences . However there are different points of concern.

The most important is the study design. The work included patients’  randomization, thus it should be a randomized clinical trial. If so, there are many methodological aspects that were not  observed. For example, a calculation of the most appropriate sample size was not carried out. As concerns randomisation, a crucial point, there is not any detail about it. Was it automated? Which system was used for treatment assignement? Was randomization stratified, for example by dialysis age or other parameters?

There is not a specific table dedicated to the description of the two groups at  baseline. Table 2 indicates also the basal values,  but the analysis is performed on the trend of the data (in three different time points) between the two groups.  Important information about patients  is missing: gender, dialysis age, body weight/body mass index (that could have an effect on the penetration of infrared light into the skin), comorbidities (especially macro-and microvascular pathology), diabetes, residual urinary oputput…  In essence, there is not any evidence that the two groups were perfectly matched for the main clinical variables . As a consequence any result is questionable.

As concerns the general conclusion about the possibility that FIR could reduce the incidence of peritonitis, this is an interesting and debatable point. Most peritonitis are caused by intestinal bacterial translocation or fluid contamination: do the Authors think that FIR may protect even from these risks? A comment is advisable.

Moreover:

  • In the Abstract (line 32), but not in the Results section, the Authors write that Albumin decreased. Table 2 instead shows, for both the groups, a small increase (from 3.68 to 3.73 gr/dl for the experimental group and from 3.56 to 3.73 g/dl for the control group): the Authors should explain
  • In the Abstract (line 36), but not in the Results section, the Authors write that Glucose increased in the experimental group due to FIR therapy. Table 2 instead shows, for the experimental group, a reduction from 105 mg/dl to 99 mg/dl: the Authors should explain
  • There is not an information about the composition of the dialysis fluid, in particular as concerns its Why the Authors  discuss about non-biocompatible fluids (line 311): which fluid did they consider?
  • Line 162: CBC and DC stay for?
  • The value of glycated hemoglobin indicates that most patients were diabetics, that probably gives reason for the poor peritoneal  efficiency: the Authors did not discuss this point  

Author Response

Thank you for your valuable opinion. We fully agree with your concern.

I revised my manuscript and changed  in line numbers.

Response to Reviewer 2 Comments

Thank you for your valuable opinion. We fully agree with your concern.

Point 1: This work is very interesting, original and probably paving the way to larger experiences. However, there are different points of concern. The most important is the study design. The work included patients’ randomization, thus it should be a randomized clinical trial. If so, there are many methodological aspects that were not observed. For example, a calculation of the most appropriate sample size was not carried out. As concerns randomisation, a crucial point, there is not any detail about it. Was it automated? Which system was used for treatment assignment? Was randomization stratified, for example by dialysis age or other parameters?

Response 1:

Thank you for your valuable opinion. We fully agree with your suggestion.

We rewrote in Line 125-149.

Point 2: There is not a specific table dedicated to the description of the two groups at baseline. Table 2 indicates also the basal values, but the analysis is performed on the trend of the data (in three different time points) between the two groups.  Important information about patients is missing: gender, dialysis age, body weight/body mass index (that could have an effect on the penetration of infrared light into the skin), comorbidities (especially macro-and microvascular pathology), diabetes, residual urinary oputput…  In essence, there is not any evidence that the two groups were perfectly matched for the main clinical variables. As a consequence, any result is questionable.

Response 2:

Thank you for your valuable opinion. We fully agree with your suggestion.

We added in Line 115-122, 124-149, 231-239. Also we added data in table 2. We added gender, dialysis age, body weight/body mass index (that could have an effect on the penetration of infrared light into the skin) in the table 2, but we did not add comorbidities, diabetes, residual urinary output. The residual urinary output concern about effect of residual renal function (RRF) in initial six months since initiating peritoneal dialysis. In our research, our study was performed during 2016/1 to 2016/10. Reviewing our patients, the shortness duration since receiving peritoneal dialysis to enrolling in this study was more than one year (the latest starting peritoneal dialysis date: 2013/9/2). Therefore, the effect of residual renal function which should be considered had a minimal bias in this study. We have measured Peritoneal Kt/V, Peritoneal weekly CCr, and e GFR in the study.

Point 3: As concerns the general conclusion about the possibility that FIR could reduce the incidence of peritonitis, this is an interesting and debatable point. Most peritonitis are caused by intestinal bacterial translocation or fluid contamination: do the Authors think that FIR may protect even from these risks? A comment is advisable.

Response 3:

Thank you for your valuable opinion. We fully agree with your suggestion.

We added in line310-313.

Point 4: Moreover:

In the Abstract (line 32), but not in the Results section, the Authors write that Albumin decreased. Table 2 instead shows, for both the groups, a small increase (from 3.68 to 3.73 gr/dl for the experimental group and from 3.56 to 3.73 g/dl for the control group): the Authors should explain

Response 4:

Thank you for your valuable opinion. We fully agree with your suggestion.

We revised in the abstract and added in discussion line 319-326.

Point 5: In the Abstract (line 36), but not in the Results section, the Authors write that Glucose increased in the experimental group due to FIR therapy. Table 2 instead shows, for the experimental group, a reduction from 105 mg/dl to 99 mg/dl: the Authors should explain

There is not an information about the composition of the dialysis fluid, in particular as concerns its Why the Authors discuss about non-biocompatible fluids (line 311): which fluid did they consider?

Response 5:

Thank you for your valuable opinion. We fully agree with your suggestion.

We added in discussion line 344-361.

Point 6: Line 162: CBC and DC stay for?

The value of glycated hemoglobin indicates that most patients were diabetics, that probably gives reason for the poor peritoneal efficiency: The Authors did not discuss this point 

Response 6:

Thank you for your valuable opinion. We fully agree with your suggestion.

We added in line 177-178 and 345-351.

Round 2

Reviewer 2 Report

You made a great work , I think the paper improved.